

# Quasi-coincident Observations of Polar Stratospheric Clouds by Ground-based Lidar and CALIOP at Concordia (Dome C, Antarctica) from 2014 to 2018

Marcel Snels[1], Francesco Colao[2], Ilir Shuli[1], Andrea Scoccione[1,3], Mauro De Muro[1,4], Michael Pitts[5], Lamont Poole[6], and Luca di Liberto[1]

[1]Istituto di Scienze dell'Atmosfera e del Clima, Via Fosso del Cavaliere 100, 00133 Roma
[2]ENEA, Via Enrico Fermi 45, 00044,Frascati, Italy
[3]Aeronautica Militare, Italy
[4]Thales Alenia Space, Rome, Italy
[5]NASA Langley Research Center, Hampton, Virginia 23681, USA
[6]Science Systems and Applications, Inc., Hampton, Virginia, 23666, USA

*Correspondence to:* Marcel Snels (m.snels@isac.cnr.it)

**Abstract.** Polar stratospheric clouds (PSCs) have been observed from 2014 to 2018 from the lidar observatory at the Antarctic Concordia station (Dome C), included as a primary station in the NDACC (Network for Detection of Atmospheric Climate Change). Many of these measurements have been performed in coincidence with overpasses of the satellite-borne CALIOP (Cloud Aerosol Lidar with Orthogonal Polarization) lidar, in order to perform a comparison in terms of PSC detection and

composition classification. Good agreement has been obtained, despite of intrinsic differences in observation geometry and data sampling. This study reports, up to our knowledge, the most extensive comparison of PSC observations by ground-based and satellite-borne lidars.

The PSCs observed by the ground-based lidar and CALIOP form a complementary and congruent dataset, and allow to study the seasonal and interannual variations of PSC occurrences at Dome C. Moreover a strong correlation with the formation tem-

perature of NAT (Nitricacidtrihydrate), $T_{NAT}$, calculated from local temperature, pressure and $H_2O$ and $HNO_3$ concentrations is shown. PSCs appear at Dome C at the beginning of June up to 26 km, and start to disappear at the second half of August, when the local temperatures start to rise above $T_{NAT}$. Rare PSC observations in September coincide with colder air masses below 18 km.

## 1 Introduction

Long term ground-based and satellite-borne lidar observations provide valuable climatological data, and allow monitoring of the state of the polar stratosphere and comparison with Chemistry Climate Models (CCMs). Several Antarctic stations have been equipped with lidar for PSC observations since the 1980s. The longest time records are from Dumont D'Urville (1989-1998, 2006-present) (Santacesaria et al., 2001; David et al., 1998, 2010) and McMurdo (1990-2010) (Adriani et al., 1992, 1995, 2004; Di Liberto et al., 2014; Snels et al., 2019). The McMurdo stratospheric lidar transferred to Concordia station at Dome





C and has been operational since 2014. Concordia station has the advantage of being on the Antarctic plateau, far from the coast and well within the polar vortex during most of the winter (see Figure 1). Meteorological conditions are in general more stable than those at the lidar stations located on the coast (McMurdo, Dumont D'Urville, Davis, Syowa, Rothera, Belgrano II) and tropospheric clouds rarely obstruct the PSC observations. South Pole station is also located far from the coast and as such

shares the advantages of Dome C, but unfortunately all three lidar systems that have been operated there (Fiocco et al., 1992; Fua et al., 1992; Simpson et al., 2005; Huang et al., 2007; Campbell and Sassen, 2008) were without a depolarization channel, and thus are limited in classifying the composition of the observed PSCs. The CALIPSO (Cloud-Aerosol Lidar and Infrared

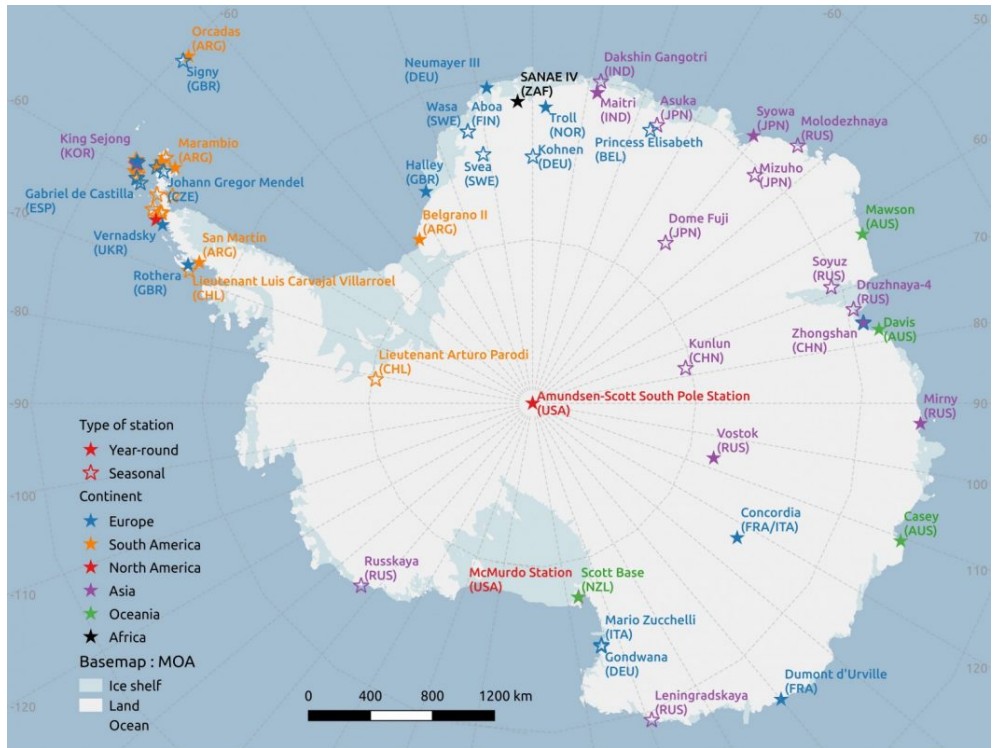

**Figure 1.** The map shows the main research stations in Antarctica.

Pathfinder Satellite Observations) satellite was launched in April 2006 as a component of the A-train satellite constellation (Stephens et al., 2002, 2017). With an orbit inclination of $98.2^{o}$, it provides extensive daily measurement coverage over the

polar regions of both hemispheres, up to $82^{o}$ in latitude. The primary instrument of CALIPSO is the Cloud Aerosol Lidar with Orthogonal Polarization (CALIOP ). CALIOP has extensively been used for observing PSCs (Pitts et al., 2009, 2011, 2013, 2018).

      The comparison of ground-based and satellite-borne lidars is useful for calibration purposes and intercomparison, but presents many difficulties, which are mainly due to the fact that perfect coincident measurements are not feasible. The rea-

son is very simple: the footprint of the satellite overpass is never exactly coincident with the position of the ground-based lidar.



Also the different observation geometries pose serious difficulties. For instance the air masses examined have different shapes; the ground-based lidar is observing an air mass determined by the divergence of the laser and the field of view of the telescope, resulting in round pixels with increasing diameter with altitude, while CALIOP produces a rectangular cross section moving along the flight track, by averaging all the laser shots along a 5 km orbit segment. Moreover, the ground-based lidar signal

suffers 2-way attenuation between the ground and detected PSCs, especially the attenuation due to tropospheric clouds, while the satellite-borne lidar is hardly attenuated down from the orbit up to the first PSCs encountered between 25 and 30 km. Then of course the operating conditions are very different; while the satellite-borne lidar is autonomously operating and acquiring data most of the time, the ground-based lidar can not be operated under severe weather conditions, and needs an operator for cleaning the view port from the snow and for trouble shooting when faults occur.

In a previous paper (Snels et al., 2019) the full dataset of PSC obervations by ground-based and satellite-borne lidar above McMurdo has been statistically compared in terms of detection and composition classification of PSCs. In the McMurdo study all ground-based lidar and CALIOP data within a longitude-latitude box ($7^o \times 1^o$) centered on McMurdo, without further constraints, were taken into account. This implies that e.g. ground-based data recorded on days without CALIOP overpasses in the longitude-latitude box were included in the statistical analysis, as well as all CALIOP profiles within the

box. Here we consider only quasi-coincident observations, which implies a much smaller set of ground-based and satellite-borne PSC observations above Concordia station, by comparing the ground-based observations with single CALIOP profiles acquired at the closest possible distance from the ground station and within 30 minutes of the ground-based observation. For the CALIPSO PSC product, a single vertical profile consists of an average over a 5 km long segment of the flight track. The quasi-coincident approach has not been applied to the McMurdo data since too few quasi-concidences were available. At

Dome C we consistently pursued to record all possible coincidences with CALIOP leading to a relatively large number of quasi-coincidences.

If we want to use ground-based and CALIOP data together, in a complementary and congruent way, we need to verify if both lidars observe essentially the same scene, in terms of PSC detection and composition. Much depends on the spatial extent of the PSC fields and their homogeneity in terms of particle composition. If the PSC's spatial extent is generally small with respect

to the average closest distances of the footprint of the satellite-borne lidar with respect to the ground station, our approach is condemned to fail. If, however, PSC fields extend over many tens or hundreds of km, our approach may yield valuable results. The position of Dome C on the central plateau, with no important orographic features present, implies that the temperature fields should be mainly of a synoptic nature, with few local perturbations. This would cause the PSCs to have a large extension and a predominantly homogeneous composition. We will show by studying PSCs observed during CALIOP overpasses close

to Dome C, that in most cases the PSC fields have sufficiently large extension to allow for a comparison of almost coincident observations. Of course this is valid for the detection, while a comparison of the composition would require that the PSCs be also of an homogeneous composition over these scales. We will also address this problem and show that in many cases the PSC composition is substantially homogenous, i.e. more than two thirds of the CALIOP overpasses in the box around Dome C at a specific vertical level have at least 75 % of the PSCs of the same composition. It should be stressed that the results presented

here are restricted to the area around Dome C, and might be very different for other locations in Antarctica.





The formation of PSCs depends on the the availability of condensation nuclei, the temperature and number densities of water and nitric acid. Nitric acid trihydrate (NAT) particles are is in thermodynamic equilibrium with the gas phase at about 6 K above the ice point (Hanson and Mauersberger, 1988), while liquid PSCs in the form of supercooled ternary solutions (STS) may form below temperatures of about 3 K above the frost point. Finally ice PSCs form below the frost point. While ice evaporates once

the temperature exceeds the frost point, NAT and STS may survive for some hours/days above their equilibrium temperature. Here we compare the occurrence of PSCs, as detected by the ground-based lidar and CALIOP, with the formation temperature of NAT. All species discussed here, NAT, STS, ice and their mixtures, form below $T_{NAT}$, and one would expect to observe PSCs when the local temperature is below $T_{NAT}$, although NAT and STS might also survive some time above this temperature. We observe how PSC occurrences drop rapidly in the third and fourth week of August, when the local temperatures exceed

$T_{NAT}$. Seasonal and interannual variations of PSC occurrences are seen to be dependent on the position of the cold polar vortex. Here we use $T_{NAT}$ as a delimiter of the area where PSCs may be formed.

## 2 PSC Observations at Dome C by Ground-based and Satellite-borne Lidar

A lidar has operated in the Antarctic station of Dome C since 2014, as a continuation of the long time series observations at McMurdo (1991-2010), with the goal to measure the atmospheric backscatter with parallel and perpendicular polarization

with respect to the emitted laser radiation. The lidar observatory uses most of the hardware previously used at McMurdo, with some small improvements. In addition, the instrument has been adapted to the harsher environment at Dome C. In particular a triple glass view port has been mounted above the telescope in order to have a better insulation from the outside temperature. Recently, the observatory has been equipped with a remote control of the laser and the data acquisition system, allowing for a complete control of the measurements from the main buildings at Dome C, at a distance of about 400 meters from the

observatory, or from our home Institute in Italy.

Dome C is well within the stratospheric polar vortex from mid-June to the end of September, although climatologically the coldest part of the vortex is migrating towards the Antarctic Peninsula starting from the second half of August. In general the weather conditions are rather stable with respect to McMurdo and other coastal lidar stations and the lidar has been operated satisfactorily from 2014 on, during the Antarctic winter, with the exception of 2019, when severe instrumental problems

occurred. The lidar is operated by science winter-overs of the PNRA (Piano Nazionale della Ricerca in Antartide) during the Antarctic winter, typically from the end of May until the end of September to cover the whole period of PSC occurrence. The lidar is operated once or twice per day, when meteorological conditions are favorable. If possible, the observations are synchronized with overpasses of the CALIPSO satellite, when its footprint is within 100 km distance from Dome C. Single vertical profiles with a vertical resolution of 60 m have been recorded by averaging 30 minutes of acquisition. All lidar data

have been deposited with the NDACC database (Snels, 2019) , where they are available to the scientific community.

CALIOP is a two wavelength lidar, measuring backscatter at wavelengths of 1064 nm and 532 nm, the latter signal separated into parallel and cross polarization, with respect to the polarization of the outgoing laser beam. Details on CALIOP can be found in (Hunt et al., 2009; Winker et al., 2009). The orbital period of CALIPSO is about 98 min, which results in about 14-15 orbits





per day. This results in two overpasses per day at distances ranging from 0 to 400 km from the ground-based lidar at Dome C, one on an ascending and the other on a descending orbit. Here we consider all overpasses within a longitude-latitude box ($7^o \times 2^o$) centered on Dome C, resulting in about 20 overpasses per month. The $7^o \times 2^o$ box corresponds roughly to a square of $200 \times 200$ km. The speed of the CALIOP footprint on the surface is about 400 km/min which implies that an overpass in the box lasts in average about 30 seconds. The orbit track of CALIOP in the box has a width of about 100 m and consists of a number of vertical profiles averaged over 5 km along the flight track.

The different sampling times and observation geometries of the ground-based lidar and CALIOP imply that it is extremely difficult to obtain "real" coincidences. An interesting approach has been suggested by David et al. (2012), who used trajectories calculated from wind velocities and directions to connect the air masses observed by ground-based lidar and CALIOP. Of course this method is limited to few intersections, and depends strongly on the altitude, since the wind velocity increases with altitude. The average windspeed ranges from about 50 to 110 km/h between 10 and 20 km of altitude, but maximum wind speeds exceed 200 km/h. The wind direction at Dome C is mostly between NE and SE. We've explored the possibility to apply the trajectory approach to our data, but the number of coincidences is very low, and not homogeneously distributed for all altitudes and this method has thus been discarded. Instead we compared ground-based data with the closest profile on each CALIPSO flight track (within 100 km) and with an overpass time within 30 minutes of the ground-based observation. This is a good compromise for obtaining a significant number of comparisons and having a reasonable probability that both lidars observe similar air masses, although not perfectly coincident. The comparison is made considering the detection and composition classification of PSC clouds.

### 2.1 PSC Detection and Classification Criteria for the CALIPSO v2 Data

The CALIOP v2 PSC detection and composition classification algorithm has been used to create the recently released CALIOP v2 PSC mask database covering the period from June 2006 to October 2019. Here we compare these v2 data with ground-based observations at Dome C from 2014 to 2018. Major enhancements in the v2 algorithm over earlier versions include daily adjustment of composition boundaries to account for effects of denitrification and dehydration, and estimates of the random uncertainties u($\beta_\perp$) and u($R$) due to shot noise in each data sample, which are used to establish dynamic detection thresholds and composition boundaries. The CALIOP v2 algorithm is represented pictorially in Figure 2 and is described in more detail in (Pitts et al., 2018; Snels et al., 2019).

### 2.2 PSC Detection and Composition Classification Criteria for the Ground-based Lidar Data

In order to compare the ground-based lidar data to the CALIOP data we have adopted a similar algorithm which follows the same approach and uses the same optical parameters as the v2 CALIOP algorithm (see Figure 2). The v2 background aerosol thresholds $\beta_{\perp,thresh}$ and R$_{thresh}$ have been calculated in a different way, to take into account a series of errors due the small deviations from the calculated molecular scattering profiles in conditions of clear sky (i.e. absence of aerosols). Then statistical errors caused by the photon counting process and thus depending on the altitude and on the possible attenuation in the lower



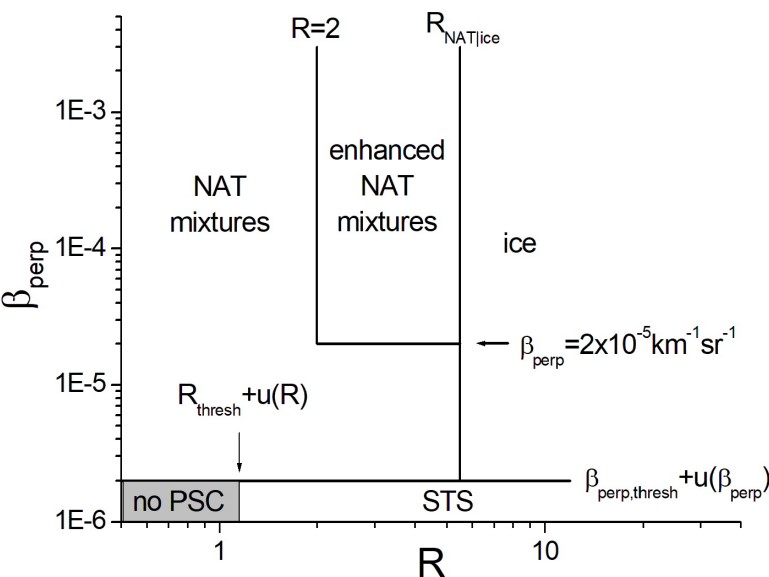

**Figure 2.** The figure shows the detection and classification criteria of the V2 CALIOP algorithm. The classification as STS, NAT mixtures, enhanced NAT mixtures and ice, requires that threshold conditions for $R$ and/or $\beta_\perp$ are satisfied. See the text for details.

troposphere, have been taken into account to calculate $u(\beta_\perp)$ and $u(R)$ and create the dynamic thresholds for detection and classification.

### 2.2.1 Data processing

Raw data consist of photon counts recorded in 400 ns bins, corresponding with a vertical resolution of 60 m and are accumulated in records with a duration of two minutes. First the data are averaged and the background count is determined from the first 40 bins, before the laser fires. The background is then subtracted from the signals. The lidar records two channels, one collecting the signal with the polarization parallel to the laser emission and the other with perpendicular polarization. The two components are separated by using two polarizing beamsplitter cubes. The perpendicular polarization is in theory due to the depolarization by molecules and aerosols. In practice some instrumental factors may contribute to the perpendicular polarization, such as a small perpendicular component of the laser emission, the residual transmission/reflection of the unwanted component of





the polarizing beamsplitter cubes, and other effects. In our case there is a substantial contribution due to the triple viewport. Starting from the lidar equation

$$S(z) = \frac{h}{2}\frac{C}{z^2}\frac{\beta(z)}{4\pi}exp[-2\int_0^z \sigma_{ext}(z')dz'] \tag{1}$$

we can express the signals on the two detectors, if we neglect the extinction, while only considering the crosstalk as originating from the optical elements (polarizer, and viewport), as

$$S_\parallel(z) = g_1\frac{1}{z^2}\beta_\parallel(z)(1-CT) \qquad \text{and} \qquad S_\perp(z) = g_2\frac{1}{z^2}(\beta_\perp(z)+CT\beta_\parallel)$$

where $g_1$ and $g_2$ are the gain factors of the two detectors and CT is the crosstalk from the parallel channel to the perpendicular channel. We neglect the crosstalk from perpendicular to parallel channel.

In order to facilitate the interpretation of the signals and the detection of clouds, we divide both signals by the molecular backscatter coefficient and multiply by $z^2$ and we get

$$r_\parallel(z) = g_1\frac{\beta_\parallel(z)(1-CT)}{\beta_{\mathrm{mol}}(z)} \qquad \text{and} \qquad r_\perp(z) = g_2\frac{(\beta_\perp(z)+CT\beta_\parallel)}{\beta_{\mathrm{mol}}(z)}$$

The molecular backscatter coefficient has been calculated by using local temperature and pressure provided by radiosoundings and, where these were not available, from NCEP.

We can normalize these expressions to 1, where no aerosols are present (typically between 26 and 30 km) The normalized expressions become

$$r'_\parallel(z) = \frac{r_\parallel(z)}{rn_\parallel} \qquad \text{and} \qquad r'_\perp(z) = \frac{r_\perp(z)}{rn_\perp}$$

where

$$rn_\parallel = g_1\frac{\beta_{\mathrm{mol}\parallel}}{\beta_{\mathrm{mol}}}(1-CT) \qquad \text{and} \qquad rn_\perp = g_2\left[\frac{\beta_{\mathrm{mol}\perp}}{\beta_{\mathrm{mol}}}+CT\frac{\beta_{\mathrm{mol}\parallel}}{\beta_{\mathrm{mol}}}\right]$$

With some algebra we can now obtain expressions for $R$ and $\beta_\perp$

$$R(z) = \frac{(1-CT)}{1+\delta_{\mathrm{mol}}}\left[r'_\parallel(z)+r'_\perp(z)\frac{\delta_{\mathrm{mol}}+CT}{(1-CT)}\right] \tag{2}$$

and

$$\beta_\perp(z) = \frac{\delta_{\mathrm{mol}}+CT}{1-\delta_{\mathrm{mol}}}\left[r'_\perp(z)-r'_\parallel(z)\frac{CT}{\delta_{\mathrm{mol}}+CT}\right]\beta_{\mathrm{mol}}(z) \tag{3}$$





where $\delta_{\mathrm{mol}} = \beta_{\mathrm{mol}\perp}/\beta_{\mathrm{mol}\|}$. In our case, using an optical bandpass filter centered at the laser wavelength (532 nm) with a FWHM of 2 nm, $\delta_{\mathrm{mol}}$ is 0.007 (Behrendt and Nakamura (2002)).

Now we can see that the crosstalk CT can be written as

$$CT = \frac{\frac{g_1}{g_2}\frac{rn_\perp}{rn_\|} - \delta_{\mathrm{mol}}}{1 + \frac{g_1}{g_2}\frac{rn_\perp}{rn_\|}} \tag{4}$$

The two parameters $rn_\|$ and $rn_\perp$ can be determined from the calibration process for aerosol free regions, so we only need the ratio of the two gain constants of the two detection channels, which can be determined e.g. by switching the detectors, or by more sophisticated methods (Snels et al. (2009)).

Now we perform the correction for extinction, using Klett and the proportionality between $\beta$ and the extinction coefficient as reported by Gobbi (1995) and use the ratios obtained after the correction to calculate $R(z)$ and $\beta_\perp(\mathrm{z})$.

**2.2.2   Error processing**

The statistical errors deriving from photon counting process, $\mathrm{u}(\beta_\perp)$ and $\mathrm{u}(R)$, have been determined from the raw signals, and are thus dependent on z. The background aerosol thresholds $\beta_{\perp,thresh}$ and $\mathrm{R}_{thresh}$ have been determined mainly by comparing with clear sky profiles and have been expressed in terms of the ratios $r'_\|(z)$ and $r'_\perp(z)$. They are estimated to be 1.1 for $r'_\|(z)$ and for $r'_\perp(z)$. For z < 16 km, these values increase gradually to 1.2 in order to take into account an insufficient correction of
saturation effects.

**2.2.3   PSC Detection and Composition Classification**

PSC detection and classification from lidar measurements with orthogonal polarization is based on two optical parameters derived from the optical signals with parallel and perpendicular polarization with respect to the laser. Here we use a method that approximately follows the v2 classification and detection scheme (see also (Snels et al., 2019)), proposed by Pitts et al.
(2018) for the classication of the CALIOP PSC data, and using the backscatter ratio $R$ and the perpendicular backscatter coefficient $\beta_\perp$.

The backscatter ratio $R$ and the perpendicular backscatter coefficient $\beta_\perp$ have been determined from the raw data as described above. The optical parameters obtained in this way, as well as their errors were smoothed to the vertical scale of the CALIOP profiles, with a vertical resolution of 180 m per layer or pixels as we will call the single bins on a profile from now
on. The detection thresholds for backscatter ratio $R$ was thus determined to be $\mathrm{R}_{thresh} + \mathrm{u}(R)$ and the threshold for $\beta_\perp$ as $\beta_{\perp,thresh} + \mathrm{u}(\beta_\perp)$ . This results in dynamic thresholds that vary from profile to profile, for instance due to attenuation of the lidar signal by cirrus clouds, and vary with altitude, mostly because of the statistical errors in the photon counting process.

In order to detect a PSC, it is sufficient that either the backscatter ratio $R$ or the perpendicular backscatter coefficient $\beta_\perp$, exceeds the respective threshold. A final step of the processing requires that at least 5 consecutive points on a vertical profile
are identified as PSCs, in order to avoid the appearance of "spikes" in the profiles. Sequences of less than five PSC points are thus considered to be non PSCs. This procedure is similar to the coherence criterium used for the CALIOP data.



### 2.2.4 PSC Composition

Composition classification for ground-based PSCs is nearly identical to the CALIOP v2 procedure, the exception being that we use values of $R_{\mathrm{NAT|ice}}$ reported for the closest profiles in the v2 CALIOP data files. The borderline value to discriminate between STS and NAT is equal to the detection threshold for $\beta_\perp$, exactly the same as for the CALIOP data.

## 3 Comparison of PSCs as Observed and Classified by Ground-based and Satellite-borne Lidar

The maximum number of overpasses within a distance of 100 km is about 20 per month, resulting in about 80 possible coincidences per PSC season, considering the observation period from June until September. However, we must consider several practical issues; sometimes the ground-based lidar cannot be operated due to adverse weather conditions, and after the end of the austral winter, the day illumination might affect the lidar measurements, reducing the maximum altitude with an
10 acceptable signal to noise ratio. Also the CALIOP instrument is subject to periods of inactivity, although very rarely.

| year | CALIOP tracks | ground based profiles | coincident profiles | detected by both lidars [%] | only detected by ground-based [%] | only detected by CALIOP [%] |
|---|---|---|---|---|---|---|
| 2014 | 91 | 99 | 26 | 71 | 23 | 6 |
| 2015 | 86 | 122 | 36 | 68 | 25 | 7 |
| 2016 | 97 | 153 | 47 | 78 | 19 | 3 |
| 2017 | 67 | 113 | 33 | 84 | 11 | 5 |
| 2018 | 89 | 137 | 30 | 76 | 12 | 12 |

**Table 1.** The number of available data and the detection statistics of both lidars has been listed. The column "detected" indicates when both ground-based lidar and CALIOP detect a PSC or they do not detect a PSC in the same vertical bin. The other two columns indicate all cases when only one of the instruments detects a PSC in a specific vertical bin. Only bins below 26 km have been considered, since very few PSCs have been observed above 26 km.

Table 1 shows some statistics illustrating the number of data acquired and actually used for comparison. Note that Table 1 includes all CALIOP tracks passing within a 200×200 km square around Dome C, including also some tracks with a closest distance of more than 100 km. This explains why the numbers of CALIOP overpasses might be slightly larger than the 80 overpasses mentioned before. Throughout this paper, the comparison for detection and composition classification will be
performed by comparing vertical bins, with the same height and a thicknes of 180 m, one of the ground-based lidar profile and the other of the nearest CALIOP profile. From now on we will refer to these bins as pixels, since each bin will be color coded in the figures in correspondence with the composition of the PSC. In our analysis we will consider only pixels between 12 and 26 km, since very few PSCs have been observed above 26 km and inclusion of the pixels above 26 km would not be a good measure for comparison.



## 3.1 Statistical analysis of CALIPSO observations in the box around Dome C for what concerns PSC extension and homogeneity of composition.

Since almost no exact coincidences are available, we compare the ground-based profiles acquired within 30 minutes from the CALIOP overpass with the closest point on the overpass. This results in an average distance between the closest points and the location of Dome C of about 50 km. The average time difference is about 15 minutes corresponding with a distance of about 15-30 km, considering the average wind speed between 10 and 20 km of altitude. Thus our approach is valid as long as the PSC clouds have an horizontal extension of about 50 km or more. To test this hypothesis we examined all CALIOP overpasses passing within the 200×200 km square around Dome C and with a minimum distance of less than 120 km. Applying these restrictions, 369 tracks for the five years (2014-2018) remain for a further analysis. The average length of each overpass track (i.e the part of the overpass within the box) was about 180 km, and 88 tracks did not contain PSCs at all. This means that for these tracks (23.8 % of the total) no PSC was detected in the CALIOP data base for any of the vertical levels from 12 to 30 km. All the altitude levels for the remaining 281 tracks were then tested to evaluate the PSC horizontal extent for all vertical levels on a 180 m grid. Now we consider each individual overpass at a specific height, and we divide the overpass in the box in 5 km segments. The CALIOP data base provides a detection flag and a composition classification for each segment. The fraction (percentage) of segments with a positive PSC detection is calculated for each overpass track. When taking into account all 281 overpasses at all heights (equal to the number of overpasses multiplied by the number of vertical "levels" of 180 m height, from 12 to 26 km) , we can obtain a statistical distribution, that indicates how often an overpass track, between 12 and 26 km, has a certain PSC fraction. The results of this procedure have been illustrated for 2016 in figure 3. It can be observed that 83.5 % of the altitude tracks have more than 80 % of positive PSC detection. This implies that for 83.5 % of the overpasses PSC clouds have approximately an horizontal extension of at least 150 km.

A second test has also been performed to quantify how much the PSC composition varies along the track. For this test we consider 4 PSC classes; STS, NAT mixtures, enhanced NAT mixtures and ice. We count the number of PSC segments along each orbit track at each altitude that belong to one of these four classes. The results are displayed in Table 2. The numbers in this Table have been obtained as follows; considering the total number of horizontal slices of all overpasses (78 slices of 180 m, from 12 to 26 km, multiplied by the number of overpasses), we count the number of slices where either STS, NAT mixtures, enhanced NAT mixtures or ice represent more than 95 % of all segments with PSCs. Dividing this number by the total number of slices we obtain the percentage of horizontal slices where one composition class is present for more than 95 %.

Table 2 shows that a fair percentage of the detected PSC layers has an almost uniform PSC composition (in average 43.6 % of all layers), but evidently about 25 % of all PSC layers has at least one third of minor species, and in some cases there might be three PSC classes on the same layer (a layer is 180 m). In conclusion, as far as the CALIOP overpasses might be considered a valid representation of the PSC population in the box around Dome C, we might say that if PSCs are present in a layer, their horizontal extension is at least of the order of 100-150 km. On the other hand, although different PSC classes might exist in the same layer, the composition is homogeneous over a substantial portion of the layer. Of course these conclusions should be interpreted with some care, since we do not have a two-dimensional PSC field at disposition, and we base our conclusions only





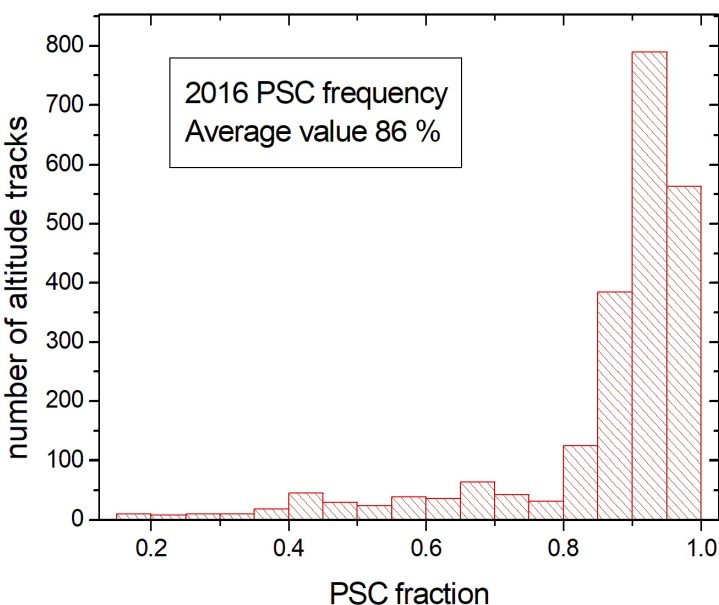

**Figure 3.** The frequency of PSC fractions has been displayed for the 2016 CALIOP data.

| PSCs of the same class | 2014 | 2015 | 2016 | 2017 | 2018 | all years |
|---|---|---|---|---|---|---|
| >95 % | 46.5 | 36.6 | 39.6 | 46.4 | 49.0 | 43.6 |
| >85 % | 56 | 51.6 | 54.6 | 59.4 | 61.7 | 56.7 |
| >75 % | 66.6 | 63 | 66 | 69.1 | 72.7 | 67.5 |
| >65 % | 78 | 73 | 70.4 | 80.4 | 82.6 | 76.9 |

**Table 2.** The percentage of pixels on the altitude tracks with at least X percent of PSCs of the same class.

on one-dimensional tracks. However, the overpasses have different directions, due to the ascending and descending orbits of CALIPSO and thus many overpasses will ultimately fill a two-dimensional field.

## 3.2   Comparison of ground-based and CALIOP data for 2015-2018.

Now that we have some confidence that a comparison between ground-based and CALIOP data at an average distance of 50 km
5   and within 30 minutes of the observation is a reasonable way of proceeding, we will illustrate the full procedure of detection and composition classification comparison with the 2015, 2016, 2017 and 2018 data, excluding 2014, being a year with fewer data.





The detection and composition classification of the PSCs in CALIOP data is based directly on the CALIOP v2 PSC Mask database, while for the ground-based data we have applied the detection and classification scheme as has been discussed above. Table 1 shows how the data reduction proceeds: from a large number of CALIOP and ground-based vertical profiles only about 25-30 % remains as we consider only coincident profiles.

Figures 4, 5, 6 and 7 (left columns) show all measured profiles for the years 2015-2018, that meet the coincidence criteria (closest CALIOP profile within 30 minutes of ground-based observation time). Please note that on several days no PSCs at all were detected in the coincident profiles. Although these "clear sky" profiles are a minority of the measurements, the two data sets agree rather well also for these clear sky profiles and thus they have been included in the detection statistics (see Table 1).

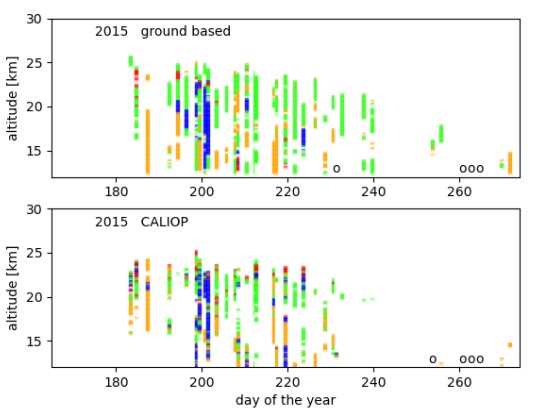
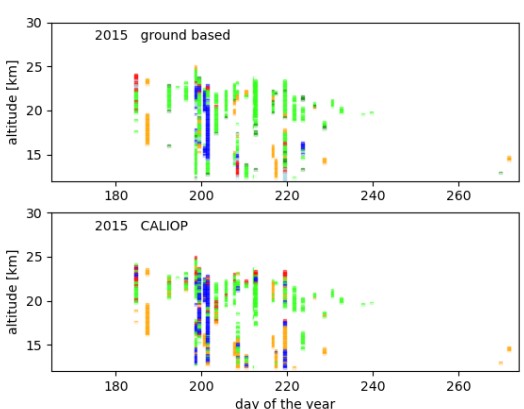

**Figure 4.** Coincident (left column) and PSC observations by both lidars (right column) for 2015 above Dome C. Upper panel: Ground-based lidar Lower panel: CALIOP above Dome C. The different colors indicate PSCs of different composition; green = NAT mixtures, yellow = STS, red= enhanced NAT mixtures, blue =ice. The circles indicate measured profiles without PSCs.

The dataset of all pixels of the coincident profiles consists of four categories of PSC detection: PSC detected by both ground-
10 based and CALIOP lidars, no PSC detected by either ground-based or CALIOP lidar, PSC detected by the CALIOP lidar only, and PSC detected by the ground-based lidar only. The first two categories are listed in the column "detected" (see Table 1), because there is agreement between the (non) detection of a PSC. There is a small fraction ($\approx 5\%$) of coincidences where only CALIOP detects a PSC, but a larger fraction (11-25 %) of coincidences where only the ground-based lidar detects a PSC. The larger fraction of coincidences with ground-based lidar only detection of PSCs may be partly due to the larger sensitivity
of the Dome C lidar below 15-16 km. This can be also seen by comparing figures of all coincident measurements and those observations where both lidars detected a PSC (left and right columns in Figures 4,5,6 and 7).

For comparison of PSC composition, we restrict the analysis to only those measurement pixels where both lidars detected a PSC (see the right columns of Figures 4, 5, 6 and 7). The results of this comparison are reported in Table 3. The first four columns report the percentage of pixels with a certain PSC composition as determined by CALIOP. The columns under the





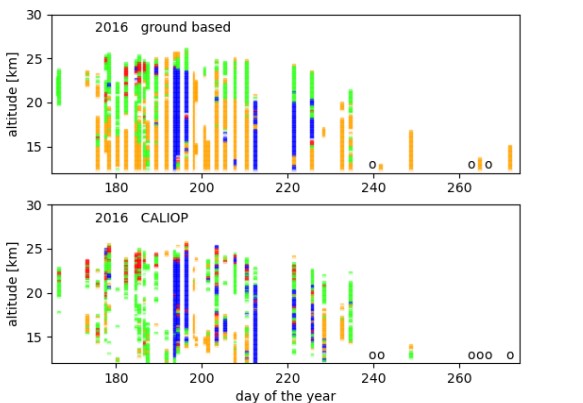
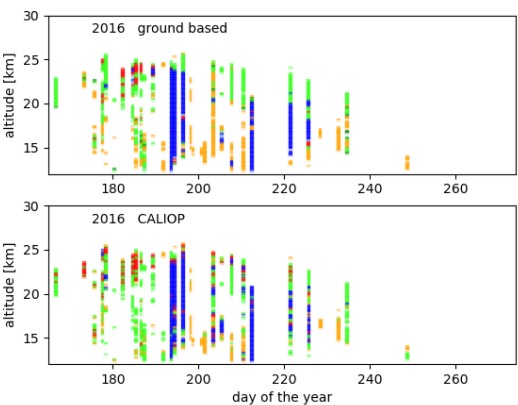

**Figure 5.** Coincident (left column) and PSC observations by both lidars (right column) for 2016 above Dome C. Upper panel: Ground-based lidar Lower panel: CALIOP above Dome C. The different colors indicate PSCs of different composition; green = NAT mixtures, yellow = STS, red= enhanced NAT mixtures, blue =ice. The circles indicate measured profiles without PSCs.

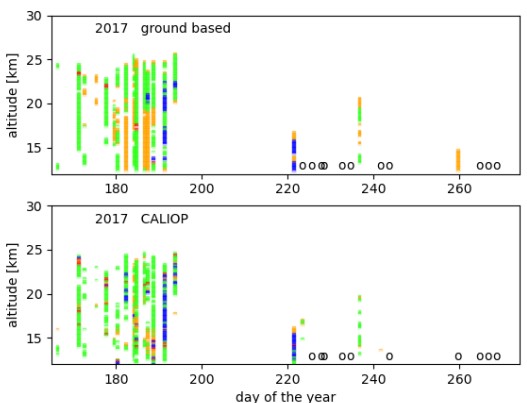
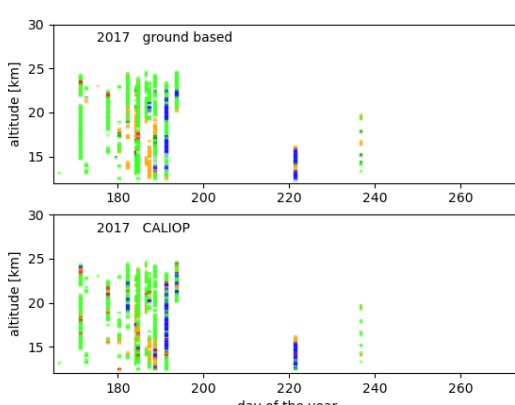

**Figure 6.** Coincident (left column) and PSC observations by both lidars (right column) for 2017 above Dome C. Upper panel: Ground-based lidar Lower panel: CALIOP above Dome C. The different colors indicate PSCs of different composition; green = NAT mixtures, yellow = STS, red= enhanced NAT mixtures, blue =ice. The circles indicate measured profiles without PSCs.

header gb method 1, report the percentage of pixels where the ground-based lidar finds the same composition for that pixel. If we exclude 2014, in average about 60 % of the ground-based pixels show the same PSC composition as reported for CALIOP. NAT mixtures show the best agreement while the enhanced NAT mixtures observed by CALIOP are very often not observed as such by the ground-based lidar.





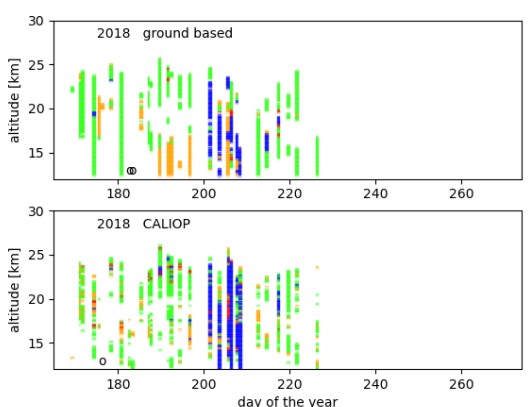
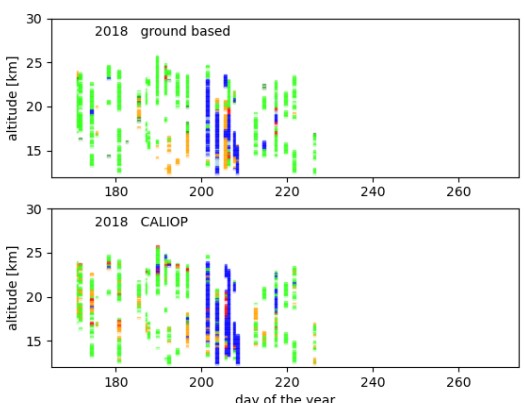

**Figure 7.** Coincident (left column) and PSC observations by both lidars (right column) for 2018 above Dome C. Upper panel: Ground-based lidar Lower panel: CALIOP above Dome C. The different colors indicate PSCs of different composition; green = NAT mixtures, yellow = STS, red= enhanced NAT mixtures, blue =ice. The circles indicate measured profiles without PSCs.

However, we want to explore how critically the PSC composition of the ground-based data depends on the choice of the thresholds, also considering that the calculated threshold for $\beta_\perp$ in the classification of the ground-based data may be affected by instrumental errors, and also $R_{\mathrm{NAT|ice}}$ may be slightly different at Dome C with respect to the value for the closest CALIOP profile. To take into account these uncertainties in both $\beta_{\perp threshold}+ \mathrm{u}(\beta_\perp)$ and $R_{\mathrm{NAT|ice}}$ we allow a 10% tolerance on both.
This implies that all PSCs with a value of $\beta_\perp$ between $\beta_{\perp threshold}+ \mathrm{u}(\beta_\perp) \pm 10\ \%$ are possibly STS or NAT and all PSCs with a value of the backscatter ratio R between $R_{\mathrm{NAT|ice}} \pm 10\ \%$ are possibly enhanced NAT mixtures or ice. This additional tolerance gives slightly better scores for the comparison (see Table 3, column gb method 2), but doesn't change the results in a significant way.

While performing ground-based measurements with a duration of several hours, we have observed that PSC layers may
move up and down on the time scale of 30 minutes. So we allowed also a small vertical displacement of the air mass observed by the ground-based lidar with respect to the CALIOP observation, which is in average at a distance of 50 km from Dome C. This implies that each CALIOP pixel is now compared with three ground-based pixels, one at the same height and the other two at the next upper or lower layer ($\pm$ 180 m) (method 3). This last method leads to an improvement of about 10 % of the overall agreement and is shown in the column gb method 3 of Table 3. The largest effect can be observed for STS and NAT
mixtures. The last three columns in Table 3 report the sum of the pixels (for each of the four composition classes), where the ground-based lidar identifies the same composition as CALIOP, for each for the three methods applied.

The results of the comparison show that in average for 58 % of all observations both lidars observe PSCs of the same composition, which becomes 71 % when tolerances on the thresholds are applied as well as on the altitude ($\pm$ 1 layer). NAT mixtures, being the dominant species, show an agreement better than 80 % , while ice and STS are slightly worse. Significantly





| | CALIOP | | | | gb method 1 | | | | gb method 2 | | | | gb method 3 | | | | tot 1 | tot 2 | tot 3 |
|---|---|---|---|---|---|---|---|---|---|---|---|---|---|---|---|---|---|---|---|
| year | STS | NAT | enh | ice | STS | NAT | enh | ice | STS | NAT | enh | ice | STS | NAT | enh | ice | all | all | all |
| 2014 | 38 | 45 | 6 | 10 | 10 | 31 | 2 | 4 | 11 | 31 | 2 | 5 | 18 | 33 | 2 | 5 | 47 | 50 | 59 |
| 2015 | 24 | 48 | 7 | 21 | 9 | 37 | 2 | 9 | 11 | 39 | 2 | 10 | 15 | 45 | 2 | 12 | 57 | 62 | 73 |
| 2016 | 18 | 42 | 11 | 27 | 13 | 21 | 3 | 22 | 13 | 22 | 3 | 22 | 16 | 28 | 4 | 24 | 59 | 61 | 72 |
| 2017 | 14 | 67 | 5 | 13 | 7 | 50 | 1 | 6 | 7 | 51 | 1 | 7 | 8 | 58 | 2 | 8 | 64 | 66 | 76 |
| 2018 | 16 | 51 | 5 | 28 | 4 | 42 | 0 | 18 | 4 | 42 | 1 | 19 | 5 | 49 | 1 | 22 | 64 | 66 | 76 |

**Table 3.** The percentage of the different PSC composition classes has been listed for CALIOP and ground-based observations following the three methods explained in the text. The last three columns are the percentages of correctly classified PSCs with respect to the CALIOP classification, according to the three methods.

fewer enhanced NAT have been observed by the ground-based lidar with respect to CALIOP. A possible explanation might be that CALIOP may better resolve smaller patches of differing composition embedded within an otherwise homogeneous PSC, since its measurements are effectively an instantaneous "snapshot" along the orbit track. On the other hand, the ground-based lidar integrates over 30 minutes and is averaging the optical parameters used for the classification, thus promoting the

NAT mixtures classification, having intermediate values for the optical parameters with respect to the minor species, the latter producing the more extreme low or high values of the optical parameters. For instance if the ground-based lidar observes STS for 10 minutes and NAT mixtures for 20 minutes, the average value of $\beta_\perp$ will be probably higher then the detection threshold and thus the observation will be classified as NAT mixtures, while CALIOP on the corresponding overpass track might identify both STS and NAT mixtures, and the closest profile represents the statistical distribution of STS and NAT mixtures. In any

case, considering the expected composition homogeneity derived from the analysis made for all CALIOP overpasses in the box around Dome C (see Table 2), the overall result is satisfactory.

## 4   PSC occurrence as Observed and Classified by Ground-based and Satellite-borne Lidar

PSC occurrence and composition classification has been performed for all ground-based data and all satellite-borne lidar profiles at the shortest distance from Dome C (thus not limited to the quasi-coincident measurements), using detection and

composition classification criteria as mentioned before. The ice frost temperature $T_{ice}$ and the formation temperature for NAT (Nitricacidtrihydrate), $T_{NAT}$, have been calculated from local temperature, pressure and $H_2O$ and $HNO_3$ concentrations. Local temperatures and pressures have been taken from ERA5, while $H_2O$ and $HNO_3$ concentrations are provided by MLS. NAT PSCs form below $T_{NAT}$, while STS occur generally about 3 degrees below $T_{NAT}$, while water ice PSCs form below the frost temperature.





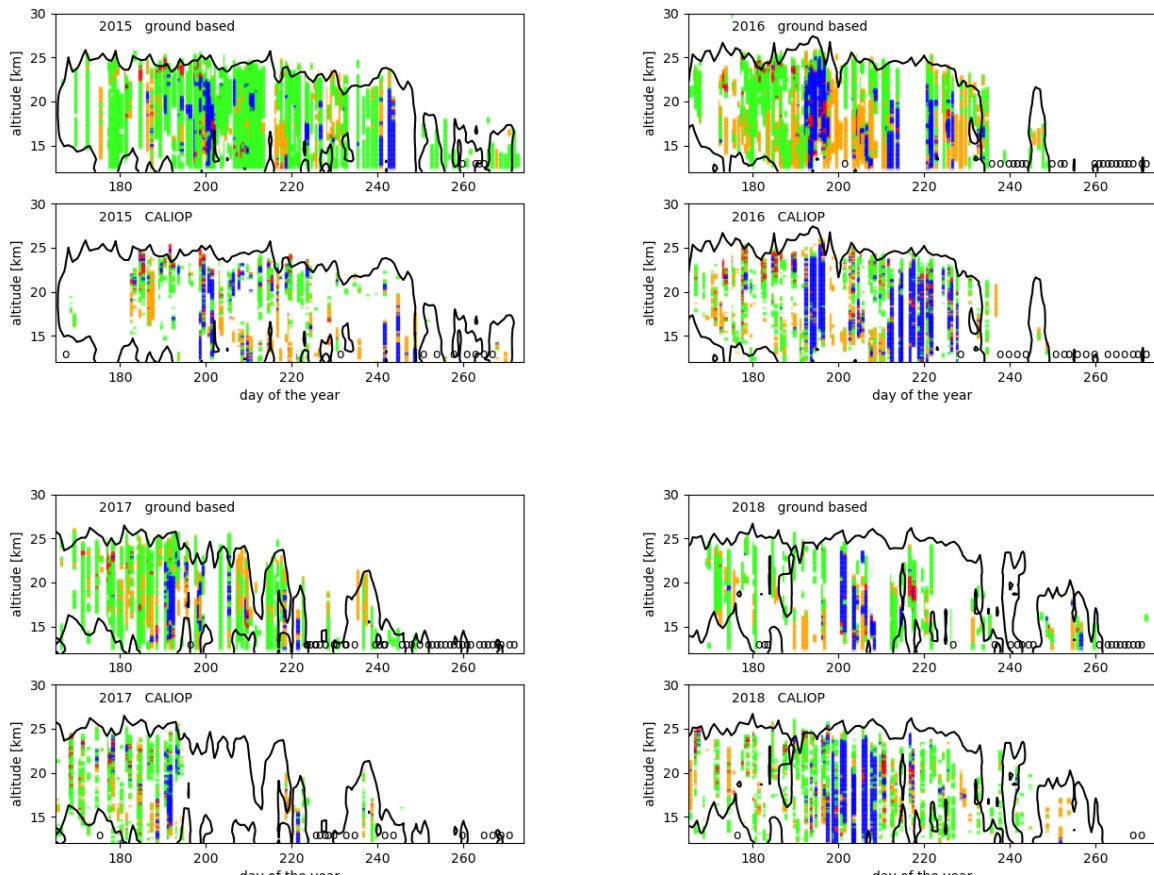

**Figure 8.** All PSC observations for 2015-2018 above Dome C. Upper panels: Ground-based lidar Lower panels: CALIOP above Dome C. The different colors indicate classified PSCs; green = NAT mixtures, yellow = STS, red= enhanced NAT mixtures, blue =ice. The circles indicate measured profiles without PSCs. The black contour indicates the area with a temperature below $T_{NAT}$.

Figure 8 shows all ground-based measurements and the closest profile of all CALIOP overpasses within 100 km and contour plots indicating where the local temperature is below $T_{NAT}$. It can be observed that PSCs are rarely observed at temperatures above $T_{NAT}$. It is also clear that the temperature being below $T_{NAT}$ is not a sufficient condition to observe PSCs. Often PSC layers are separated by layers where neither of the two lidars observes PSCs. It is remarkable that the highest altitude where PSCs have been observed by the lidars coincides almost perfectly with the $T_{NAT}$ contour, which implies that the sensitivity of the ground-based lidar at Concordia station is generally not a limiting factor for the observations at high altitudes. It also evident that from the second half of August Dome C is outside the coldest part of the vortex. This implies that ice and STS PSCs are seldomly formed, while NAT occurrences are few. Looking at the previous figures, one can notice that CALIOP hardly observes any PSCs in September, while the ground-based lidar, being more sensitive due to the better S/N ratio of the



lidar signals, observes mostly NAT. In 2015 the vortex is weakening only at the beginning of September, while in 2017 already around the $10^{th}$ of August the local temperature between 12 and 30 km exceeds $T_{NAT}$. Ice PSCs are mainly observed during the last three weeks of July, although a minor event can be observed at the end of August in 2015 and a important occurrence in the second week of August in 2016. We did not find evidence of a strong correlation with $T_{ice}$ contours. For all years very

few PSCs have been observed in September. While the polar vortex has an almost circular symmetry around the South pole during the winter, a displacement of the cold pool versus the Antarctic peninsula has been observed, starting from September and sometimes from the second half of August. Pitts et al. (2018) calculated twelve-year (2006-2017) monthly means f PSC occurrence frequency at 20 km in the southern hemisphere and showed that these correlate strongly with contour maps of $T_{\mathrm{NAT}}$ and $T_{\mathrm{ice}}$.

## 5  Conclusions

Lidar measurements of PSCs from the Dome C lidar observatory have been compared in terms of detection and composition with the data obtained with the space-borne lidar CALIOP. In order to elaborate the ground-based lidar data, detection and composition classification algorithms have been developed, using criteria very similar to those used for the CALIOP data, taking into account systematic and statistical errors. Since exact coincidences are practically non-existent, quasi coincident

measurements have been defined as being recorded within 30 minutes of the CALIOP overpass and at the nearest distance from Dome C on each overpass, but in any case within 100 km. This implies a severe reduction of the available data, but resulted anyway in a significant number of comparisons for every year. The validity of such a comparison, with quasi coincidences, depends strongly on the uniformity of the PSC fields in the Dome C area. The hypothesis of sufficiently large horizontal extensions of the PSC clouds close to Dome C has been tested by considering all overpasses within a square of $200 \times 200$

km centered on Dome C. A large number of overpasses with PSC detection resulted to be almost contiguous within the 200 $\times$ 200 km square, This suggests that uniform PSC fields predominate in the area around Dome C. The detected PSC clouds might consist of PSCs of different composition (here we consider STS, NAT mixtures, enhanced NAT mixtures and ice), interleaved both horizontally and vertically, and although the NAT mixtures are the dominant class during most of the winter, small patches of the minor species might be present. By studying the CALIOP composition classification along all overpasses

at single altitude levels, it appeared that nearly half of all overpasses with PSCs showed contiguous layers of a single PSC class, while for about 25 % of the layers at least one third of minor species was found. This implies that while about half of the horizontal PSC fields have a homogeneous composition, a non neglectable part evidences the presence of PSCs with different composition within the Dome C area ($200 \times 200$ km). As a result we feel confident that a comparison between quasi coincident measurements is fully justified for PSC detection and to a lesser extent for the composition classification. The comparisons are

based on five years of data, from 2014 to 2018, and comprise 172 quasi coincident vertical profiles.

The result of the detection comparison is that about 75 % of the (non) PSCs were detected by both lidars, while about 5 % was detected only by CALIOP and 20 % only by the ground-based lidar. Probably the better detection efficiency of the ground-based lidar at lower altitudes might explain the latter. If we consider only 2016, 2017 and 2018 these values are even better and





reach 76 to 84 % of agreement. The composition of the detected PSCs has been compared in a strict, pixel-to-pixel way, and also by introducing some more permissive criteria, such as a small variation of the classification thresholds and a comparison with the next higher or lower layer ($\pm$ 180 m). It can be concluded that the observation of NAT mixtures by CALIOP is confirmed in most cases (83 %) by the ground-based lidar, while the identification of the minor species by CALIOP, was confirmed in

average for 59 , 32 and 67 % of the cases, for STS, enhanced NAT mixtures and ice, respectively, by the ground-based lidar. Our explanation is that the ground-based data acquisition produces averaged values of the optical constants, by integrating over 30 minutes, corresponding with a spatial integration of 15-30 km. This integration process favours the classification as NAT mixtures, at the cost of reducing classification of the other species. On the other hand CALIOP takes a "snapshot" during its overpass of about 30 seconds, and is more sensitive to the other species.

The results presented here are providing a solid basis for the comparison of ground-based and space-borne lidar observations of polar stratospheric clouds (PSC). This is the most extensive comparison of such data, to our knowledge, and may provide a means to produce a standard PSC product for ground-based lidars, with a good compatibility with CALIOP and other space-borne instruments, although with some caveats. It opens new possibilities of including ground-based validated PSC data in CCM models and microphysical studies. The method proposed here is shown to be valid for polar regions with rather

uniform temperature fields, and absence of important orographic structure, but might be used with some constraints to different situations.

It has also been shown that observations obtained by the ground-based lidar and CALIOP are complementary and congruent and can be used to study seasonal and inerannual variations of the presence of PSC clouds at Dome C. The PSCs observed by both systems are generally observed for local temperatures below $T_{NAT}$, although some observations at higher temperature are

reported. These are mostly NAT mixtures, that are known to persist some days even above $T_{NAT}$. During the winter season, PSCs slowly descend and are rarely observed from the second half of August, in agreement with the warming of the vortex at Dome C.

For all five years concerned here, few PSCs have been observed during the second half of August and during September, most probably because a displacement of the cold pool versus the Antarctic pensinsula. Presently a climatological study for the

Dome C area is underway by combining data of both lidars, which, based on this study, are compatible up to a large degree.

## 6   Data availability

The raw data of the ground-based lidar at McMurdo are publicly available on the NDACC data base

(ftp://ftp.cpc.ncep.noaa.gov/ndacc/station/mcmurdo/ames/lidar/). The CALIOP v2 data are available on request from Michael Pitts and Lamont Poole.





## 7   Author contributions

MS was responsible for most of the writing, review and editing process, supported by all co-authors, with major contributions from FC, MP and LP. The lidar data analysis was conducted by MS with contributions from AS. MdM installed the lidar at Dome C. FC , MdM and LdL performed the calibration and maintainance of the ground-based lidar. MP and LP provided the CALIOP data and discussed the interpretation of the lidar data in many occasions.

## 8   Competing interests

The authors declare that they have no conflict of interest.



*Acknowledgements.* The authors acknowledge the financial support by PNRA in the frame work of the projects 2009/B.08 and OSS-12. We also acknowledge the support of the ISSI-PSC initiative project. Logistical and winter-time technical support was provided by the Piano Nazionale della Ricerca in Antartide (PNRA). The authors thank Igor Petenko, Giampietro Casasanta, Simonetta Montaguti, Alfonso Ferrone, Filippo Cali Quaglia and Meganne Christian for performing the ground-based lidar measurements at Dome C during the winter.



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
