# Peer review of "Quasi-coincident Observations of Polar Stratospheric Clouds by Ground-based Lidar and CALIOP at Concordia (Dome C, Antarctica) from 2014 to 2018"

_Atmospheric Chemistry and Physics, 2020_

## Referee Comment (RC1) · Anonymous Referee #1 · 3 Nov 2020

Snels et al. use lidar observations made from Concordia for the time period 2014-2018 and provide a quasi-coincident data set to the space-borne measurements by CALIOP on CALIPSO. They show that this lidar dataset is complementary and congruent to the CALIPSO data and can thus be used to study the seasonal and inter-annual variation of PSCs at Concordia.

This study is definitely useful for the scientific community and does deserve to be published. However, I have some major points of criticism that should be consid-

ered before publication.

**General comments:**
1) What is the goal of this study? To provide the coincident data set or studying the seasonal and inter-annual variations of the PSCs? The former has been definitely done, but not the latter. This has only been scratched at the surface. The differences should be discussed in more detail. This leads me directly to my second point.

2) The present version of this manuscript is quite technical and would have made a better fit in AMT or ESSD. To adjust the manuscript to the ACP standards the scientific content of this study should be elaborated in more detail.

3) The method applied in this study is not entirely clear, e.g. how has the extension of the PSCs determined? How has it been determined that lidar and CALIOP measured the same air mass? The time and distance criteria is here to my opinion not enough; one would need an additional criteria. Where has it been shown or documented that Concordia is well within the polar vortex during most of the winter?

4) The differences in the analyses and lessons learned between Snels et al. (2019) and the present study should be pointed out more clearly.

5) In my opinion the structure of the paper would be more logical if first the agreement in occurrence and extension of the PSCs would be discussed before discussing the measured PSC types. Additionally, a comparison as presented in Figure 4 in Snels et al. (2019) would also be for Concordia quite valuable.

**Specific comments:**
P1, L6: It may be correct that your study is the first study reporting an extensive comparison between ground-based and space-borne lidars. However, there have

been other studies before e.g. for the Arctic by Achtert et al., (2011). This should be considered and discussed. Of course not in the abstract, but in the main text of the manuscript. See my comments below.

P2, Figure 1: It would be quite helpful if the location of Concordia could be emphasized in the figure. You could e.g. add a colored box around Concordia.

P3, L21: What is the advantage of having quasi-coincident measurements? What is the difference between the data set derived for McMurdo in Snels et al. (2019) and the one derived for Concordia. Are both data sets equally valuable for the scientific community? This also should be discussed in more detail.

P3, L31: How has this been studied? How has the extension of the PSCs been determined?

P3, L33: Add a discussion on the different schemes. Achtert and Tesche (2014) provide such a comparison. Although this comparison was made for the Arctic, the derived results are also valid for the Antarctic. I remember that Pitts et al. improved their scheme based on the discrepancies found in the Achtert and Tesche (2014) study and that should be discussed here.

P5, L9: Also here, though on the Arctic, Achtert et al. (2011) is a good example.

P5, L15: To have the profiles within in 100 km distance and 30 min time difference is no guarantee for being in the same air mass. An additional criteria is needed, as e.g. PV or temperature.

P6, Figure 2: This is the same figure as in Snels et al. (2019). This should be
at least mentioned in the caption. However, it would also be enough to skip this figure and just refer to Snels et al. (2019). For the threshold values anyway a table would be much more helpful. Especially, the differences between CALIOP and the Concordia lidar thresholds should become clearer.

P6, L2: This should help to overcome the problems found in Achtert and Tesche (2014). Has a comparison been made to check if the results (for single profiles) between CALIOP and lidar really agree?

P6, L3-P8, L16: Isn't that a standard procedure for processing lidar data. Thus, isn't the whole section obsolete since this is documented elsewhere? This part could be put in an appendix or a supplement.

P9, L3: It is not entirely clear what value is used here. Wouldn't it be easier to show the agreement between CALIOP and lidar on one example PSC?

P9, Table 1: Why is still such a high amount of PSC detected by only one of the instruments?

P10, L7: Not clear if here all CALIPSO overpasses have been used or only the ones where a PSC was detected.

P10, L8ff: Not clear what has been done.

P10, L20: I cannot follow this line of reasoning. These paragraphs need definitely to be improved to understand what actually has been done.

P10, L20: How can you be sure that this is the same PSC and not another PSC?

P10. L27: Also here I have difficulties to follow. Please improve the text so that it becomes clearer what you have done.

P11, Figure 3: How does this figure look for the other years?

P11, Table 2: How does this table look for the lidar data?

P12, L1: Differences between the characterization schemes could be pointed out more clearly.

P12, Figure 4: Please add a legend. Not clear what is shown. Should not the left panels show the data and the right panels the coincident data?

P12, L13: The uncertainties and disagreements between the data sets should be discussed in more detail. Why does CALIOP or the ground-based lidar detect PSCs that the other instruments does not detect?

P13, Figure 5: Same as for Figure 4, please add a legend and check if the orders of panels agree with what is written in the caption.

P13, Figure 6: Same here as for Figure 4 and Figure 5.

P13: To my opinion it would be easier and more logical to first compare the PSC occurrence and extension before comparing the PSC composition.

P14, L10: This shift between ground-based lidar and CALIOP is nothing new. This has already been documented in Achtert et al. (2011).

P15, L1: Also here the Achtert and Tesche (2014) paper could be helpful for the discussion.

P16, Figure 8: The caption could be clearer on what is actually shown in which panel. For example that the always two panels are for one year and then all year from 2015-2018 are shown could be more clearly stated. Also here a legend to the figures would be more helpful then listing the color coding up in the caption.

P17, L9: The differences between the PSC seasons of the different years should be discussed in more detail.

P17, L19: Tested by what? What was the criteria?

P18, L9: Couldn't also here the results of the Achtert and Tesche (2014) paper be helpful for the discussion?

P18, L10ff: The recent study by Tesche et al. (2020) in ACPD could also be of interest for the discussion of the results in this study.

**Technical corrections:**
P4, L3: ice point → ice frost point

P4, L4: frost point → ice frost point

P4, L6: Here we compare → Here, we compare

P4, L25: I know what you mean, but I think "science winter-overs" is not the correct expression.

P4, L33: Citation should be embedded in the text, thus "(Hunt et al., 2009, Winker et al., 2009)" should be changed to "Hunt et al. (2009) and Winker et al. (2009)".

P5, L26: Same here with the citation of Pitts and Snels.

P8, L11: deriving → derived

P8, L11: from photo counting → from the photo counting

P15, L15: The ice frost → The ice frost point

P16, L8: means f PSC → means of PSC

P22, L14: montly → monthly

P22, L16: Models → models

P22, L21: 0, null → obsolete ?

**References:**
Achtert, P., and M. Tesche (2014), Assessing lidar-based classification schemes for polar stratospheric clouds based on 16 years of measurements at Esrange, Sweden, J. Geophys. Res. Atmos., 119, doi:10.1002/2013JD020355.

Achtert, P., F. Khosrawi, U. Blum, and K. H. Fricke (2011), Investigation of polar stratospheric clouds in January 2008 by means of ground-based and spaceborne lidar measurements and microphysical box model simulations, J. Geophys. Res., 116, D07201, doi:10.1029/2010JD014803.

Tesche, M., Achtert, P., and Pitts, M. C. (2020): Location controls the findings of ground-based PSC observations, Atmos. Chem. Phys. Discuss., https://doi.org/10.5194/acp-2020-930, in review.
* * *
Interactive
comment

---

## Referee Comment (RC2) · Anonymous Referee #2 · 3 Dec 2020

GENERAL SUMMARY AND COMMENTS

This paper presents an analysis of approximately coincident polar stratospheric cloud (PSC) observations between ground-based lidar measurements from a single location in Antarctica and CALIOP satellite lidar measurements. The current work is a refinement of previous work (by the same lead author) that considered a larger combined dataset at a different Antarctic location, which used only geographic coincidence as a requirement. Specific temporal and spatial coincidence requirements are developed to maximize the consistency of the scene between the ground-based and satellite instruments. While these requirements reduce the number of candidate events that can be analyzed, analysis indicates that a majority of events identify PSCs with the same composition. These results suggest that appropriately filtered ground-based PSC measurement data sets can be constructed for further comparisons and model studies.

This paper is generally well-written and constructed. However, I have some requests for clarification before recommending it for publication.

TECHNICAL COMMENTS

1. p. 8, lines 29-30: This requirement means that any PSC detections considered in the analysis must have a minimum thickness of 0.9 km. Do you find any problem with situations where there may be a gap in a cloud layer, e.g. 4 PSC points followed by 1 non-PSC point followed by 3 PSC points, that would cause a potential PSC detection to be discarded?

2. p. 10, lines 18-20: Is this result evaluated for a single altitude along each track (e.g. 17.28 km), or does the PSC altitude vary along any given track? If the PSC altitude changes by 1-2 km, with perhaps a corresponding change in temperature, then possibly the composition changes along the track. This question does seem to be addressed in the next two paragraphs.

3. p. 11, lines 6-7: Table 1 shows 26 coincident profiles in 2014, compared to 30 profiles in 2018 and 33 profiles in 2016. This small difference in number doesn't seem like a strong reason to exclude the 2014 season.

4. p. 13, line 1: Please clearly state that 'gb' represents "ground-based" here to avoid confusion.

5. p. 15, lines 16-17: Is there a reason for using ERA5 temperature and pressure data here vs. NCEP temperature and pressure data previously (p. 7, lines 12-13)? The differences are probably small, but clarification would be helpful.

TYPOGRAPHICAL ERRORS

p. 1, line 8: "allow to" should be "allow us to".

p. 9, line 15: "thicknes" should be "thickness".

p. 16, line 6: "It also" should be "It is also".

p. 17, line 12: "elaborate" could be "evaluate".

p. 17, line 27: "neglectable" could be "negligible".
* * *

---

## Author Comment (AC1) · 23 Dec 2020

Answers to referee 1 We thank the referee for dedicating his precious time to read and comment the manuscript.

Snels et al. use lidar observations made from Concordia for the time period 2014-2018 and provide a quasi-coincident data set to the space-borne measurements by CALIOP

on CALIPSO. They show that this lidar dataset is complementary and congruent to the CALIPSO data and can thus be used to study the seasonal and inter-annual variation of PSCs at Concordia. This study is definitely useful for the scientific community and does deserve to be published. However, I have some major points of criticism that should be considered before publication. General comments: 1) What is the goal of this study? To provide the coincident data set or studying the seasonal and inter-annual variations of the PSCs? The former has been definitely done, but not the latter. This has only been scratched at the surface. The differences should be discussed in more detail. This leads me directly to my second point.

Answer: The goal of the study is twofold. Ground-based and satellite-borne lidars are complementary and can provide useful data for climate studies. It would be desirable if the ground-based data would be representative for a certain area around its location and could be used to fill gaps in time between CALIOP overpasses. It is well known that the PSC detection and classification obtained by different ground-based and satellite borne lidars depends on several parameters and causes biases when comparing ground-based data versus CALIOP. The main reasons can be summarized as follows: 1) Different observation geometry. The airmasses observed by gb and CALIOP are different in size- The S/N ratio has a different dependence on altitude. For the gb lidars the S/N ratio decreases with altitude, while the S/N ratio for CALIOP has only a small dependence on altitude (between 12 and 30 km). The quality of the data obtained by gb lidars depends on the tropospheric cloud cover (See paper Tesche) 2) Different detection and classification thresholds (see Tesche and Achtert). Gb lidars often use fixed thresholds, CALIOP in its v2 algorithm uses dynamical thresholds 3) Different duration for the acquisition of a vertical profile. GB lidars usually integrate over 30-60 minutes, CALIOP takes "snapshots" with the duration of several seconds. 4) It is almost impossible to sample the same airmass with gb and CALIOP. There practically no perfect coincidences Here we try to reduce the effect of 2) by applying exactly the same detection and classification algorithm and by using dynamical thresholds. We also mitigate 4) by comparing quasi-coincident data and by making a statistical study of the

continuity of PSCs in terms of detection and classification in the box around Dome C. We then discuss the impact of 1) and 3) on the resulting detection and classification of the quasi-coincident PSCs. To our opinion this is not a merely technical study, but has scientific merits, since it deals with real data, probably the largest dataset of PSC data available presently and it provides a better way to integrate gb and CALIOP data also for future studies.

The difference with the previous paper on the McMurdo PSC observations is evident. For McMurdo we didn't have sufficient quasi-coincident data to do this and we simply took all McMurdo observations (also on days when there were no CALIOP overpasses) and all CALIOP profiles in a box around McMurdo. This may give some statistical agreement, even if there is no day to day agreement. Suppose CALIOP observes NAT for three days when the ground-based lidar is not observing, and then the ground-based lidar observes NAT for three days when there are no close CALIOP overpasses, we have an overall agreement, but this doesn't mean we have agreement on single profiles. It would be ideal if we could study exact coincidences in time and space, but these opportunities are very rare. The approach of using trajectories like proposed by David et al. and Achtert et al, is valid, but again limited to case studies.

2) The present version of this manuscript is quite technical and would have made a better fit in AMT or ESSD. To adjust the manuscript to the ACP standards the scientific content of this study should be elaborated in more detail.

Answer: We tried to better explain how we obtain the dynamical thresholds and how we study the continuity of PSCs in the box around Dome C. To our opinion this paper is not mainly technical since it addresses problems of possible biases between different lidar systems, and tries to eliminate some of them, while estimating the impact of others. The discussion of seasonal and interannual variability in the 5 years of observations at Dome C is just an example of how lidar data from different instruments can be used together. We are preparing a much more detailed study covering a longer period and data of different lidars. This study will use the results of the present paper.

3) The method applied in this study is not entirely clear, e.g. how has the extension of the PSCs determined?

Answer: The extension of the PSCs has been inferred from the many CALIOP over-passes in the box around Dome C. This is described in section 3.1 "Statistical analysis of CALIPSO observations in the box around Dome C for what concerns PSC extension and homogeneity of composition." The statistical analysis is performed by considering the sequence of pixels at the same height on the same overpass. (A pixel is a point on a vertical profile. Each pixel represents a volume of 180 m x 100 m x 5000 m(height x horizontal swath of the CALIOP track x distance between profiles on the overpass track))The number of pixels with a positive detection for PSC with respect to the total number of pixels on the overpass track in the box is a measure of the continuity of the cloud. In the same way, when comparing the classification of each pixel, we get a measure for the homogeneity of the PSC clouds in terms of the PSC species.

How has it been determined that lidar and CALIOP measured the same air mass? The time and distance criteria is here to my opinion not enough; one would need an additional criteria.

Answer: We don't say that the two lidars measure the same airmass. We just assume that the two lidars measure PSCs in a reasonable homogeneous cloud with a sufficient horizontal extension.

Where has it been shown or documented that Concordia is well within the polar vortex during most of the winter?

Answer: Several definitions exist in literature for the determination of the borders of the polar vortex. The commonly accepted one use the potential vorticity (PV) to determine the edge of the vortex. Waugh et al (IAS 1999) describe in detail the extension and the position of the polar vortex of the SH wrt to the South Pole (ref). It shows that Dome C is within the polar vortex from May through September, at potential temperatures between 500 and 800 K.

4) The differences in the analyses and lessons learned between Snels et al. (2019) and the present study should be pointed out more clearly. 5) In my opinion the structure of the paper would be more logical if first the agreement in occurrence and extension of the PSCs would be discussed before discussing the measured PSC types. Additionally, a comparison as presented in Figure 4 in Snels et al. (2019) would also be for Concordia quite valuable.

Answer: We thank the referee for this observation. We've added a paragraph in the manuscript to better explain the scope of this study and the difference with the previous one. Also we try to distinguish between the impact of the main differences between gb and CALIOP observations in general. While in the McMurdo paper it made some sense to produce a statistics of the PSC composition vs the local temperature, since a large part of the data was not coincident, so the ground-based and CALIOP lidars might observe the same species (e.g. STS) in different conditions. Here we have a different situation, since we compare two observations of PSCs at approximately the same coordinates and at the same time, i.e. with the same temperatures.

Specific comments: P1, L6: It may be correct that your study is the first study reporting an extensive comparison between ground-based and space-borne lidars. However, there have been other studies before e.g. for the Arctic by Achtert et al., (2011). This should be considered and discussed. Of course not in the abstract, but in the main text of the manuscript. See my comments below.

Answer: We thank the referee for drawing attention to the Achtert and Tesche paper and included it in our discussion.

P2, Figure 1: It would be quite helpful if the location of Concordia could be emphasized in the figure. You could e.g. add a colored box around Concordia.

Answer: This has been done.

P3, L21: What is the advantage of having quasi-coincident measurements? What is

the difference between the data set derived for McMurdo in Snels et al. (2019) and the one derived for Concordia. Are both data sets equally valuable for the scientific community? This also should be discussed in more detail.

Answer: See the answers above. We added a paragraph discussing this.

P3, L31: How has this been studied? How has the extension of the PSCs been determined?

Answer: See above. See section 3.1

P3, L33: Add a discussion on the different schemes. Achtert and Tesche (2014) provide such a comparison. Although this comparison was made for the Arctic, the derived results are also valid for the Antarctic. I remember that Pitts et al. improved their scheme based on the discrepancies found in the Achtert and Tesche (2014) study and that should be discussed here.

Answer: Here we don't want to discuss the merits of the different schemes for detection and composition classification of lidar data. We want to remove the differences that typically occur by applying different schemes, as has been explained so well in Achtert and Tesche (2014), in order to have a minimum bias due to the applied scheme. Since we compare with CALIOP v2 data, we try to approximate the v2 scheme, by applying dynamic thresholds also for the ground-based lidar data.

P5, L9: Also here, though on the Arctic, Achtert et al. (2011) is a good example.

Answer: We inserted the reference to the work of Achtert et al.

P5, L15: To have the profiles within in 100 km distance and 30 min time difference is no guarantee for being in the same air mass. An additional criteria is needed, as e.g. PV or temperature.

Answer: Of course the probability that both lidars observe the same air mass is extremely low. We argue that the extension of the PSC fields make it plausible that both

lidars observe PSCs that are part of the same large PSC field ( or cloud cover if you prefer). It is not necessary that they observe the same air mass at different times. Additional criteria would reduce the available data enormously as for the method of David or suggested by Achtert (2011).

P6, Figure 2: This is the same figure as in Snels et al. (2019). This should be at least mentioned in the caption. However, it would also be enough to skip this figure and just refer to Snels et al. (2019). For the threshold values anyway a table would be much more helpful. Especially, the differences between CALIOP and the Concordia lidar thresholds should become clearer.

Answer. We added in the figure capture that it is the same as in Snels(2019). The figure doesn't indicate absolute thresholds ( apart from that between enhanced NAT and NAT mixtures), but is much more clear than any Table would be.

P6, L2: This should help to overcome the problems found in Achtert and Tesche (2014). Has a comparison been made to check if the results (for single profiles) between CALIOP and lidar really agree?

Answer: We do better than comparing single profiles, we compare many profiles for 5 years of measurements (see Figures 4-7). Comparing one single profile is not statistically significant. Comparing many profiles shows how good (or bad) the comparison really is.

P6, L3-P8, L16: Isn't that a standard procedure for processing lidar data. Thus, isn't the whole section obsolete since this is documented elsewhere? This part could be put in an appendix or a supplement.

Answer: This is not a standard procedure for lidar as far as we know. This procedure has not been documented elsewhere, so we prefer to keep it for further reference and also to make the procedure from raw data to the optical parameters which have been used to make the detection and composition classification transparent.

P9, L3: It is not entirely clear what value is used here. Wouldn't it be easier to show the agreement between CALIOP and lidar on one example PSC?

Answer: Here we don't refer to a single value. The threshold values are dynamic, and may change from profile to profile and also as a function of the altitude.

P9, Table 1: Why is still such a high amount of PSC detected by only one of the instruments?

Answer: There are essentially three reasons. The first is that the ground-based lidar has in average a lower detection threshold; we found that Rthreshold has an average value of 1.08 for the ground-based lidar and 1.15 for CALIOP. This major sensitivity accounts for about 5 to 10 % of the PSCs detected only by the ground-based lidar. The second is that we are not sure that the air masses observed by both lidars are part of the same large cloud. We expect that this does not occur frequently, since we have shown that around Dome C quite large PSC extension may be expected. The third is that a "hole" in the cloud deck is detected with a higher probability by CALIOP, since the observation time for a profile is only a few seconds, while the ground-based lidar integrates for about 30 minutes and thus observes a displacement of the cloud deck due to the wind. The frequency of holes in the cloud deck can be estimated from the analysis we performed on the CALIOP overpasses; about 10 % of all overpasses where PSCs were observed had a partial cloud cover (i.e along the track profiles with PSCs were observed as well as profiles without any PSC).

P10, L7: Not clear if here all CALIPSO overpasses have been used or only the ones where a PSC was detected.

Answer: The text says "all CALIPSO overpasses"

P10, L8ff: Not clear what has been done.

Answer: We tried to explain better in the text.

P10, L20: I cannot follow this line of reasoning. These paragraphs need definitely to

be improved to understand what actually has been done.

Answer: We tried to explain better in the text.

P10, L20: How can you be sure that this is the same PSC and not another PSC?

Answer: As stated before we can not be completely sure that both lidars observe different parts of the same cloud. But the probability that they do is rather high. (see Figure 3)

P10. L27: Also here I have difficulties to follow. Please improve the text so that it becomes clearer what you have done.

Answer: We tried to explain better in the text.

P11, Figure 3: How does this figure look for the other years?

Answer: The figures for other years are similar.

P11, Table 2: How does this table look for the lidar data?

Answer: The Table has been obtained from CALIOP overpasses and make a statistics of the number of pixels (a single level on a profile) where a PSC was detected wrt to the total number of pixels for that overpass-level. Obviously a similar analysis can not be made for the ground-based lidar

P12, L1: Differences between the characterization schemes could be pointed out more clearly.

Answer: Essentially both schemes follow the same principles, but the thresholds have been determined in a different way, due to the different nature of the data. We tried to explain this better.

P12, Figure 4: Please add a legend. Not clear what is shown. Should not the left panels show the data and the right panels the coincident data?

Answer: We agree that the legend is not completely clear. On the left we have the

quasi-coincident profiles, on the right we have the same profiles but we kept only those pixels where both lidars detect a PSC (thus eliminating the pixels where only one of the two lidars observes a PSC)

P12, L13: The uncertainties and disagreements between the data sets should be discussed in more detail. Why does CALIOP or the ground-based lidar detect PSCs that the other instruments does not detect?

Answer: See above.

P13, Figure 5: Same as for Figure 4, please add a legend and check if the orders of panels agree with what is written in the caption.

Answer: The left columns shows the ground-based (upper panel) and the corresponding quasi-coincident CALIOP data (lower panel). The right columns show only the data where both lidars observes a PSC, that is eliminating all data where only one of the lidars observes a PSC.

P13, Figure 6: Same here as for Figure 4 and Figure 5.

Answer: see above

P13: To my opinion it would be easier and more logical to first compare the PSC occurrence and extension before comparing the PSC composition.

Answer: That is exactly what we are doing: the left column shows the occurrence for ground-based (upper panel) and CALIOP (lower panel), the right column shows the composition for all pixels where both lidars observe a PSC. We do not understand how the extension can be shown, we estimated the extension only from CALIOP data to check if it makes sense to compare profiles at a certain distance from each other.

P14, L10: This shift between ground-based lidar and CALIOP is nothing new. This has already been documented in Achtert et al. (2011).

Answer. We thank the referee for drawing our attention to Achtert et al (2011). They

observed a shift of the cloud base of 1 km on a time scale of 20 hours approximately. Please note that this is not a shift wrt CALIOP but a shift of the cloud in time ! We added this in the manuscript

P15, L1: Also here the Achtert and Tesche (2014) paper could be helpful for the discussion.

Answer: The Achtert and Tesche paper deals mainly with the effect of the choice of different parameters and algorithms on the classification of the same PSC. Here we tried to eliminate most of the differences in parameters and algorithm used for ground-based and CALIOP data. So the differences in detection and composition are mainly due to the different S/N and to the fact that we compare quasi-coincident measurements that might not always observe the same PSC, as well as the difference in the duration of the observation.

P16, Figure 8: The caption could be clearer on what is actually shown in which panel. For example that the always two panels are for one year and then all year from 2015-2018 are shown could be more clearly stated. Also here a legend to the figures would be more helpful then listing the color coding up in the caption.

Answer: We indicated in each panel the year and the lidar (ground-based vs CALIOP). Each pair (ground-based vs CALIOP) is clearly grouped together for each year. So we think that there should be no reason for confusion or misunderstanding.

P17, L9: The differences between the PSC seasons of the different years should be discussed in more detail.

Answer: We added some line in the discussion

P17, L19: Tested by what? What was the criteria?

Answer: The extension of the clouds was inferred by examining ALL CALIOP tracks in a box around Dome C (see page 10 and 11).

P18, L9: Couldn't also here the results of the Achtert and Tesche (2014) paper be helpful for the discussion?

Answer: The Achtert and Tesche paper is mainly dealing with the comparison of different detection and classification schemes, which are applied to a set of lidar data in the Arctic. Here (page 18) we discuss the differences of the data acquisition of CALIOP and ground-based lidar.

P18, L10ff: The recent study by Tesche et al. (2020) in ACPD could also be of interest for the discussion of the results in this study.

Answer: The recent study by Tesche et.al. has been cited in the revised manuscript for what concerns the fact that Dome C is a considered a favourable location for PSC observation.

Technical corrections: P4, L3: ice point ! ice frost point done

P4, L4: frost point ! ice frost point Changed throughout the manuscript

P4, L6: Here we compare ! Here, we compare done

P4, L25: I know what you mean, but I think "science winter-overs" is not the correct expression. We have changed in winter-over scientists

P4, L33: Citation should be embedded in the text, thus "(Hunt et al., 2009, Winker et al., 2009)" should be changed to "Hunt et al. (2009) and Winker et al. (2009)". done

P5, L26: Same here with the citation of Pitts and Snels. done

P8, L11: deriving ! derived

P8, L11: from photo counting ! from the photo counting done

P15, L15: The ice frost ! The ice frost point done

P16, L8: means f PSC ! means of PSC done

P22, L14: montly ! monthly done

P22, L16: Models ! models done

P22, L21: 0, null ! obsolete ? corrected

References: Achtert, P., and M. Tesche (2014), Assessing lidar-based classification schemes for polar stratospheric clouds based on 16 years of measurements at Esrange, Sweden, J. Geophys. Res. Atmos., 119, doi:10.1002/2013JD020355. Achtert, P., F. Khosrawi, U. Blum, and K. H. Fricke (2011), Investigation of polar stratospheric clouds in January 2008 by means of ground-based and spaceborne lidar measurements and microphysical box model simulations, J. Geophys. Res., 116, D07201, doi:10.1029/2010JD014803. Tesche, M., Achtert, P., and Pitts, M. C. (2020): Location controls the findings of ground-based PSC observations, Atmospheric Chemistry and Physics Discussions, 2020, 1–19, doi:10.5194/acp-2020-930, https://acp.copernicus.org/preprints/acp-2020-930/, 2020.
* * *
[Figure]

**Fig. 1.** corrected figure 1

[Figure]

---

## Author Comment (AC2) · 23 Dec 2020

Answers to referee 2

We thank the referee for dedicating his precious time to read and comment the manuscript.

[Figure]

TECHNICAL COMMENTS

1. p. 8, lines 29-30: This requirement means that any PSC detections considered in the analysis must have a minimum thickness of 0.9 km. Do you find any problem with situations where there may be a gap in a cloud layer, e.g. 4 PSC points followed by 1 non-PSC point followed by 3 PSC points, that would cause a potential PSC detection to be discarded?

Answer: Yes, we tried to adopt a similar criterium of continuity as used for CALIOP. We varied the number of contiguous points from 2-5 and found that a number of 5 effectively eliminates obvious spikes above 25 km, and eliminates a negligible number of points below 25 km.

2. p. 10, lines 18-20: Is this result evaluated for a single altitude along each track (e.g. 17.28 km), or does the PSC altitude vary along any given track? If the PSC altitude changes by 1-2 km, with perhaps a corresponding change in temperature, then possibly the composition changes along the track. This question does seem to be addressed in the next two paragraphs.

Answer: Yes, we use single altitudes for making the statistics. The possibility that PSC clouds might change altitude in the box around Dome C is addressed on page 14, lines 8-14, when we compare the ground-based data with CALIOP.

3. p. 11, lines 6-7: Table 1 shows 26 coincident profiles in 2014, compared to 30 profiles in 2018 and 33 profiles in 2016. This small difference in number doesn't seem like a strong reason to exclude the 2014 season.

Answer: Yes, 2014 has not been represented in the figures, but has been included in the analysis, as can been seen in Table 3. Note that 2014 was our first season at Dome C, and data acquisition started quite late in the winter (after 13 July) , which results in less data.

4. p. 13, line 1: Please clearly state that 'gb' represents "ground-based" here to avoid

confusion.

Answer: We thank the referee for this suggestion and changed gb into ground-based in the Table and in the text.

5. p. 15, lines 16-17: Is there a reason for using ERA5 temperature and pressure data here vs. NCEP temperature and pressure data previously (p. 7, lines 12-13)? The differences are probably small, but clarification would be helpful.

Answer: ERA5 is a reanalysis with a better resolution and more vertical levels, so we preferred It to NCEP, although the differences are really small. The molecular density needed for calculating the molecular scattering, was mainly based on the local radio soundings, integrated with NCEP where necessary.

TYPOGRAPHICAL ERRORS p. 1, line 8: "allow to" should be "allow us to".

p. 9, line 15: "thicknes" should be "thickness".

p. 16, line 6: "It also" should be "It is also".

p. 17, line 12: "elaborate" could be "evaluate".

p. 17, line 27: "neglectable" could be "negligible".

Answer: we corrected all typographical errors as suggested by the referee